# Discovery of atomic clock-like spin defects in simple oxides from first principles

Joel Davidsson [1] ✉, Mykyta Onizhuk [2] ✉, Christian Vorwerk[2] & Giulia Galli [3,4] ✉

Virtually noiseless due to the scarcity of spinful nuclei in the lattice, simple oxides hold promise as hosts of solid-state spin qubits. However, no suitable spin defect has yet been found in these systems. Using high-throughput first-principles calculations, we predict spin defects in calcium oxide with electronic properties remarkably similar to those of the NV center in diamond. These defects are charged complexes where a dopant atom − Sb, Bi, or I − occupies the volume vacated by adjacent cation and anion vacancies. The predicted zero phonon line shows that the Bi complex emits in the telecommunication range, and the computed many-body energy levels suggest a viable optical cycle required for qubit initialization. Notably, the high-spin nucleus of each dopant strongly couples to the electron spin, leading to many controllable quantum levels and the emergence of atomic clock-like transitions that are well protected from environmental noise. Specifically, the Hanh-echo coherence time increases beyond seconds at the clock-like transition in the defect with [209]Bi. Our results pave the way to designing quantum states with long coherence times in simple oxides, making them attractive platforms for quantum technologies.

Point defects in semiconductors and insulators, and their associated electron and nuclear spins, are key components of quantum information systems[1,2]. In the last two decades, several defects and host crystals have been proposed[1], which are suitable for quantum technologies. Notable examples are the NV center in diamond[3–10] and the divacancy in silicon carbide[11–14], that exhibit excellent coherence properties for quantum sensing and communication. However, the search and engineering of spin defects in solids that can combine *multiple* quantum functionalities, such as computation, communication, and sensing are still open challenges.

Among promising classes of materials for the implementation of different quantum modalities are oxides and chalcogenides. Recent theoretical predictions[15] suggest that spin defects in simple oxides, such as MgO and CaO, should have nuclear spin-limited coherence times at least ten times longer than those measured in naturally

abundant diamond and SiC for the NV center and the divacancy, respectively. Importantly, in oxides, quantum coherence properties could be engineered by interfacing them with magnetic, strain, and electric fields, providing a broad parameter space over which optimization of desired functionalities may be carried out. In addition, interfacing oxides with semiconductors offers the possibility of engineering hybrid quantum optoelectronic systems, as well as the flexibility of tuning materials properties, e.g. with specific strain fields.

An essential prerequisite to designing and engineering quantum platforms using oxides is the availability of spin defects with the desired electronic properties, in addition to long coherence times. The discovery and prediction of such defects is clearly a challenging task, given the immensely vast parameter space to explore.

Here, we focus on a specific oxide, calcium oxide (CaO) that has the potential to host spin defects with Hahn-echo coherence time ($T_2$)

[1]Department of Physics, Chemistry and Biology, Linköping University, SE-581 83 Linköping, Sweden. [2]Pritzker School of Molecular Engineering, University of Chicago, Chicago, IL 60637, USA. [3]Pritzker School of Molecular Engineering and Department of Chemistry, University of Chicago, Chicago, IL 60637, USA. [4]Materials Science Division and Center for Molecular Engineering, Argonne National Laboratory, Lemont, IL 60439, USA. ✉e-mail: joel.davidsson@liu.se; onizhuk@uchicago.edu; gagalli@uchicago.edu

of 34 ms[15] i.e. 30–40 times longer than in natural SiC or diamond, respectively. CaO contains minimal nuclei with non-zero spins in the naturally abundant material, and it was also identified as a promising candidate for spin qubits in a recent search of inorganic materials[16], although suitable spin defects have not yet been predicted.

In this paper, we use a high-throughput method[17,18] and calculations based on density functional theory, quantum embedding theory[19,20], and spin dynamics calculations using the cluster-correlation expansion method (CCE)[21]. Using this combination of techniques, we predict point defects in CaO with electronic properties remarkably similar to those of the NV center in diamond. In addition, these defects emit in the telecommunication range and exhibit improved coherence times due to the presence of clock transitions. Our results pave the way to engineering quantum states with long coherence times in simple oxides.

## Results

We adopt a high-throughput method that has been successfully applied to the search of spin defects in SiC[22,23], and we use the software Automatic Defect Analysis and Qualification (ADAQ)[17,18] to generate

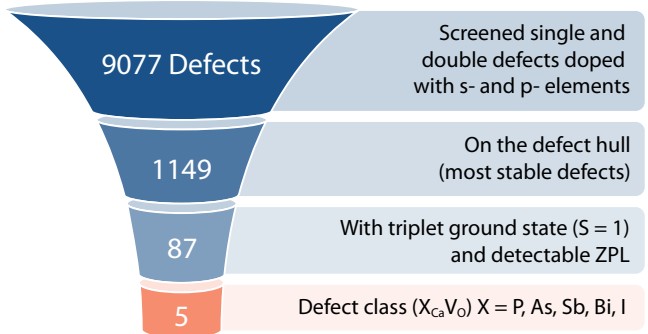

**Fig. 1 | High-throughput search.** The selection process of point defects in calcium oxide adopted in the high-throughput search used in this work (see text). We identified five defects denoted as $X_{Ca}V_O$ (bottom part of the figure) with detectable zero-phonon line (ZPL) and electronic structures similar to that of the NV center in diamond, where the dopant X = P, As, Sb, Bi, I is located between adjacent Ca and O vacancies.

single (vacancies, substitutionals, and interstitials) and double defects (such as vacancy-substitutional clusters) in CaO, a simple oxide with the rocksalt structure. We consider all non-radioactive elements in the s- and p-block of the periodic table (details in the Methodology Section). Our strategy is shown in Fig. 1. We screen a total of 9077 defects using DFT at the generalized gradient corrected (GGA) level of theory[24], of which 1149 are found on the defect hull (these defects exhibit the lowest formation energy per stoichiometry and for a given Fermi energy[22,23]). Within the set of most stable defects, we search for those with a triplet ground state (S = 1), and we identify 200 candidates; of these, 87 exhibit at least one occupied and one unoccupied localized defect state in the band gap, indicating that a zero phonon line (ZPL) should be detectable experimentally. Finally, a careful investigation of these 87 candidates reveals a class of five stable defects ($X_{Ca}V_O$), which we call NV-like, with an electronic structure similar to that of the NV center in diamond. They consist of a Schottky defect, specifically an oxygen $V_O$ and adjacent calcium vacancy $V_{Ca}$, and of a dopant atom X belonging either to group 15 or group 17 of the periodic table: X = P, As, Sb, Bi, I. For X belonging to group 15, the NV-like defects are negatively charged, $X_{Ca}V_O^-$ and, as shown in Fig. 2a, the equilibrium position of the dopant is along the shortest path connecting the Ca and O vacancies, closer to the position of the missing Ca atom. For X = P and As, we find two local minima along the $V_{Ca}$ and $V_O$ path, but only the configuration closest to the Ca site, which has the lowest energy, exhibits the desired electronic structure. However, we could not obtain a satisfactory convergence of the excited states for P and As (see Supplementary Note 1 for more details), hence in the following, we only focus on Sb and Bi. Interestingly, the $N_{Ca}V_O^-$ (the direct analog to the NV center), with a triplet ground state, does not have the desired electronic structure since N occupies the O site ($V_{Ca}N_O$). For dopants of group 17, we find that only the positively charged $I_{Ca}V_O$ complex is stable and has the same electronic structure as that of the group 15 dopants, with I located in a similar position as Sb and Bi.

The geometrical configuration of the NV-like defects identified here has $C_{4v}$ symmetry (see Fig. 2) and gives rise to four states within the band gap in both spin channels. One is close to the valence band maximum; the other three are mid-gap states. For example, for X from group 15, the mid-gap states originate from the single substitutional $X_{O^-}$ that has $O_h$ symmetry and a threefold degenerate state ($T_{1u}$). As mentioned above, the most stable position of the X dopant in $X_{Ca}V_O$, in

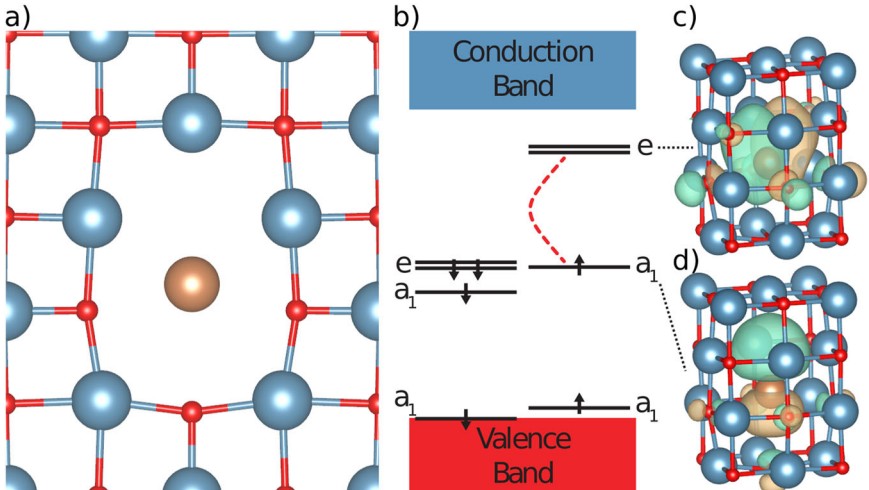

**Fig. 2 | Structural and electronic properties of predicted defects. a** Atomic configuration of the $X_{Ca}V_O$ defects identified in our search (see Fig. 1), where X = Sb, Bi, I is located between the missing cation and anion sites. For X = Sb or Bi, the defect is negatively charged; for X = I, it is positively charged. All defects have $C_{4v}$ symmetry. Red, blue and brown spheres denote oxygen, calcium and dopant atoms, respectively. **b** Electronic structure of the $X_{Ca}V_O$ complexes, where we have indicated the zero-phonon line excitation between the $a_1$ and $e$ states. States are labeled following the representation of the $C_{4v}$ group. **c** and **d** show the iso-surfaces of the sum of $e_x$ and $e_y$ defect orbitals, and of the $a_1$ defect orbital, respectively, both with a value of $10^{-4}$ Å$^{3/2}$.

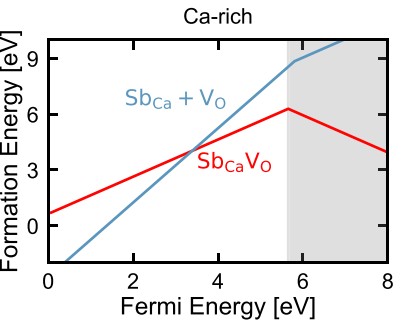
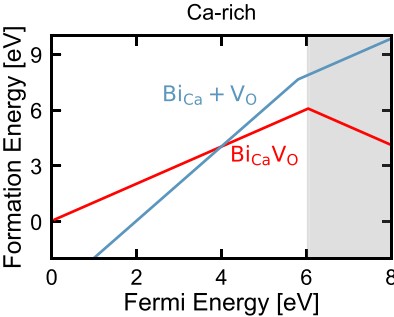
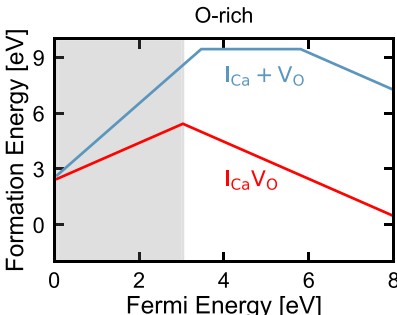

**Fig. 3 | Formation energies of predicted defects.** Each panel shows the formation energy for the $X_{Ca}V_O$ defects where X = Sb, Bi, I, obtained at the HSE level of theory with 62.5% short-range Hartree-Fock exchange (for PBE and HSE results, see Supplementary Fig. 2). Red lines are the formation energy for the complexes. $Sb_{Ca}V_O^-$, $Bi_{Ca}V_O^-$, and $I_{Ca}V_O^+$ are stable for values of the Fermi level indicated by the shaded regions and have the electronic structure shown in Fig. 2b). The blue lines are the formation energy for the separated single defects ($X_{Ca} + V_O$). When the red line is below the blue, the complex has a positive binding energy. The Sb and Bi defects require Ca-rich conditions to be stable in the negatively charged state, whereas the I defect requires O-rich conditions (see Supplementary Note 3 for details).

the absence of the adjacent Ca, is between the Ca and O vacancy sites (Fig. 2a). This geometrical configuration lowers the $O_h$ symmetry of the complex to $C_{4v}$, leading to a split of the $T_{1u}$ into $a_1$ and $e$ states; as shown in Fig. 2c), these states are highly localized.

Figure 3 shows the formation energy of the $X_{Ca}V_O$ complex and of the separate $X_{Ca} + V_O$ defects with X = Sb, Bi, I, as obtained with the HSE functional with a mixing parameter ($\alpha$) of 62.5%. The mixing parameter was chosen to reproduce the experimentally measured band gap of CaO (details are reported in the Supplementary Note 2 and 3, together with a comparison of results obtained at the PBE and HSE level of theory). The shaded regions of Fig. 3 indicate the values of the Fermi level ($E_F$) for which NV-like defects are stable and correspond to n-type conditions for Sb and Bi and p-type conditions for I. In CaO, n-type conditions could be obtained, for example, by substituting Mo with Ca[25]. The Fermi level could also be adjusted by using a Schottky diode as, e.g., in deep-level transient spectroscopy[26]. However, the Fermi level should be adjusted in a range where single intrinsic defects are stable. We investigated the range of stability of intrinsic defects to determine which growth conditions (Ca- or O-rich) are most suitable for X = Sb, Bi, I (see Supplementary Note 3). We found Ca-rich conditions are required to stabilize the Sb and Bi defects in the negative charge state, which has spin-1. Instead O-rich conditions are required for the I defect to be stable in the positive charge that has spin-1. Under appropriate growth conditions, we predict similar formation energies for $X_{Ca}V_O$ in CaO as for the NV center in diamond[27]. Our results indicate that if X is implanted in CaO at high T, where Schottky defects are expected to be present, NV-like defects should be formed upon annealing since the $X_{Ca}V_O$ complex is more stable than the separate $X_{Ca}$ and $V_O$ point defects.

We now turn to discuss the magneto-optical properties of the NV-like defects identified above, starting with the ZPL, where the excitation of interest is between the $a_1$ and $e$ states, as shown in Fig. 2b). We find that the $Bi_{Ca}V_O^-$ defect has a ZPL in the telecommunication range, and $I_{Ca}V_O^+$ has a ZPL close to the same range.

The band gap of CaO (7.09 eV[28]) is severely underestimated when using the PBE functional (see Table 1) and moderately so with the hybrid functionals HSE and K-PBE0. However, remarkably, we find approximately the same ZPL results with all functionals, except in the case of the I dopant (where differences are nevertheless within 10%). These results indicate that in an ionic material such as CaO, Coulombic interactions are dominant in determining the ZPL, and the exchange-correlation interactions have a minor effect on total energy differences.

Interestingly, we observe minor variations among the functionals for the computed geometries in the ground and excited states and the zero-field splitting. The sum of the ionic displacements between the ground and excited state ($\Delta R$) increases when using the HSE and K-PBE0 functional instead of PBE. Not unexpectedly, the same trend is seen for the zero-field splitting (ZFS). The geometry difference also affects single-particle orbitals that, in turn, affect the value of the computed dipole-dipole term of the ZFS.

To further characterize the excitations of NV-like defects, we determined the singlet-triplet (S-T) splitting for the most promising complex, emitting in the telecommunication range: $Bi_{Ca}V_O^-$. Due to the strong correlation between the localized defect orbitals, the S-T splitting cannot be described using mean field theories, such as DFT. Therefore, we employed the quantum defect embedding theory (QDET)[19], which has been shown to yield accurate results for several spin defects in wide band-gap semiconductors[10,20]. Our results, displayed in Fig. 4, confirm that the ground state of the defect is a triplet (S = 1), as obtained with DFT. The lowest triplet excitation ($^3E$) is found at 1.45 eV above the ground state ($^3A_2$). This energy is similar to the DFT absorption energy (varying between 1.2 and 1.3 eV, depending on the functional). Within the computed many-body states, we identify three singlet (S = 0) excitations. Two nearly degenerate singlet states ($^1B_1$ and $^1B_2$) occur at 0.26 eV and 0.33 eV above the ground state, while a non-degenerate singlet state ($^1A_1$) is at 0.56 eV. This many-body level diagram resembles that of the NV center in diamond[27], where two singlet excited states occur between the triplet ground state and the first triplet excited state (see Fig. 4). Since both defects have $^3A_2$ ground

**Table 1 | Computed properties of CaO and the $X_{Ca}V_O$ defects, with X=Sb, Bi, I, as obtained with three different density functionals: PBE, HSE and K-PBE0**

| Host/Defect | Property | Functional | | |
|---|---|---|---|---|
| | | **PBE** | **HSE** | **K-PBE0** |
| **CaO** | **Band gap [eV]** | **3.64** | **5.32** | **6.44** |
| $Sb_{Ca}V_O^-$ | ZPL [eV] | 0.54 | 0.53 | 0.53 |
| | $\Delta R$ [Å] | 0.49 | 0.52 | 0.53 |
| | ZFS [GHz] | 2.51 | 2.91 | 2.96 |
| $Bi_{Ca}V_O^-$ | ZPL [eV] | 0.75 | 0.76 | 0.77 |
| | $\Delta R$ [Å] | 0.49 | 0.50 | 0.51 |
| | ZFS [GHz] | 2.18 | 2.46 | 2.49 |
| $I_{Ca}V_O^+$ | ZPL [eV] | 0.79 | 0.78 | 0.69 |
| | $\Delta R$ [Å] | 0.44 | 0.60 | 0.64 |
| | ZFS [GHz] | 2.96 | 3.42 | 3.48 |

We show the band gap, the zero-phonon lines (ZPL), the sum of the ionic displacements between the ground and excited state ($\Delta R$), and the zero-field splitting (ZFS).

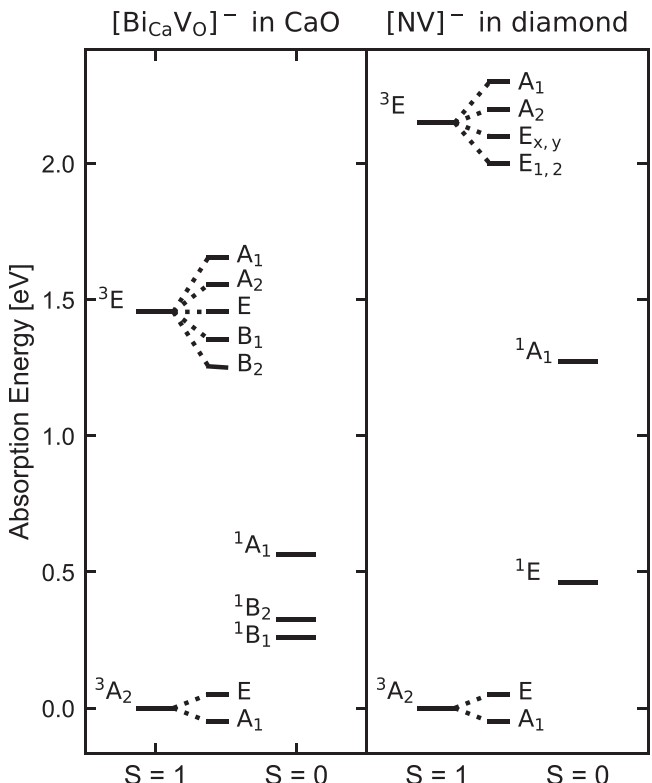

**Fig. 4 | Many-body states from quantum embedding.** Many-body state diagram for the $Bi_{Ca}V_O^-$ defect in CaO (left) compared to that of the NV$^-$ defect in diamond (right). Excitation energies are obtained using the quantum defect embedding theory (QDET). The splitting of the $^3A_2$ and $^3E$ states is due to ZFS and spin-orbit coupling, respectively. The diamond results are taken from ref. 19 and the symmetries from ref. 27.

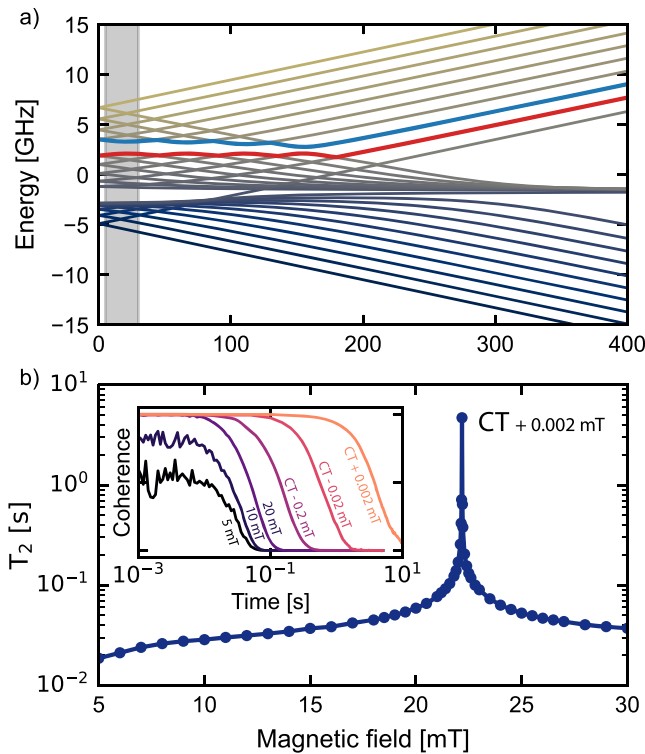

**Fig. 5 | Spin dynamics calculations. a** Spin energy levels of the $Bi_{Ca}V_O^-$ defect. Levels chosen as qubit levels are marked with red and blue. Gray shaded area represents the range of magnetic fields shown in **b**. **b** Nuclear spin-limited $T_2$ of the electron spin near a clock transition (CT), as computed using the CCE method. The inset shows actual computed coherence signal near a clock transition.

states and similar ZFS value, they both split into $A_1$ and $E$ states. To estimate the spin-orbit interaction of the $^3E$ states, we assume that the interaction parameters of $Bi_{Ca}V_O^-$ in CaO are similar in sign and magnitude as that of the NV center[27], and with this hypothesis, which remains to be verified, we find the order reported in Fig. 4. For the NV center, an optical cycle populates the $m_s = 0$ instead of the $m_s = \pm 1$ states, due to inter-system crossing mediated by phonons and spin-orbit interaction[27]. Our results indicate that there should be such a cycle for the NV-like defects in CaO as well.

As mentioned in the introduction, unlike diamond, CaO is an almost noiseless host of spin-defects, removing the need for any isotopic engineering. Natural-abundant CaO contains only about 0.13% of magnetic nuclei $^{43}Ca$ with spin-$7/2$ and about 0.04% of spin-$5/2$ $^{17}O$. As a result, the nuclear-spin limited Hahn-echo $T_2$ of the localized electron spin in CaO is 34 ms, an order of magnitude higher than of naturally abundant diamond (0.89 ms)[15]. A significant additional advantage of the defect centers discovered here is that each of them contains a *single* nuclear spin that strongly couples to the electron spin of the defect. For example, $^{209}Bi$ is a spin-$9/2$ particle with nearly 100% natural abundance. The parallel component of the hyperfine coupling between the electron and Bi nuclear spins, 1.27 GHz, is similar to Bi donors in Si[29]. Hence, the combined electron-nuclear system exhibits 30 energy levels (see Fig. 5a) that are separately addressable in experiments, providing a broad space of spin states accessible for the design of quantum technologies.

The strong electron-nuclear spin coupling in $Bi_{Ca}V_O^-$ leads to a set of avoided crossings between energy levels as a function of the magnetic field. The spin transitions between these levels, known as "clock transitions" (CT)[29], are remarkably robust to external

perturbations, and thus the coherence time of qubits operating at a CT can be substantially increased[30,31]. Using the cluster-correlation expansion method (CCE), implemented in the PyCCE code[21], we computed the coherence of the spin qubit $Bi_{Ca}V_O^-$ near CTs (see Fig. 5b). We find that at the magnetic field of 22.18 mT (2 μT from a clock transition), the $T_2$ is already increased by two orders of magnitude (4.7 s) compared to that of a qubit operating away from CTs (34 ms).

## Discussion

Table 2 summarizes the predicted physical properties of the NV center in diamond and those of the NV-like defects in CaO, for which we choose a specific functional, HSE, that has yielded results in good agreement with experiments for diamond. In CaO, the electron spin is predicted to have longer Hahn-echo $T_2$ (34 ms) than in naturally abundant diamond (0.89 ms)[15]. Furthermore, the refractive index of CaO (1.84) is lower than that of diamond (2.42) and closer to the refractive index of optical fibers (1.44), thus enabling an increase in the number of emitted photons into the fiber, if the two materials are integrated.

The computed ZPL for the NV center is 2.00 eV, which is in close agreement with the experimental value of 1.945 eV[32], and, as discussed above, those of the NV-like defects in CaO are in the range 0.53–0.78 eV. The ZPL polarization is perpendicular to the axis of the defect (if the defect is aligned with the z-axis, the transition dipole moment is in the xy-plane). Due to the symmetry of the host, the $X_{Ca}V_O$ defects can be oriented in three different directions, indicating that in experiments a similar polarization will be measured in all directions. The radiative lifetimes are comparable between NV centers and NV-like defects, all in the ns range. Note, that non-radiative effects were not considered in our study. The computed NV center radiative lifetime

**Table 2 | Comparison between computed properties of the NV center in diamond and the $X_{Ca}V_O$ defects in CaO, with X=Sb, Bi, and I, as obtained using the HSE functional**

| Host | Diamond | CaO | | |
|---|---|---|---|---|
| $T_2$ time[15] [ms] | 0.89 | 34 | | |
| Refractive index | 2.42 | 1.84[60] | | |
| Defect | NV⁻ | $Sb_{Ca}V_O{}^-$ | $Bi_{Ca}V_O{}^-$ | $I_{Ca}V_O{}^+$ |
| Symmetry | $C_{3v}$ | $C_{4v}$ | $C_{4v}$ | $C_{4v}$ |
| ZPL [eV] | 2.00 | 0.53 | 0.76 | 0.78 |
| ZPL [nm] | 619 | 2338 | 1624 | 1593 |
| TDM [Debye²] | 48 | 1205 | 118 | 1748 |
| Radiative lifetime [ns] | 6.5 | 18 | 63 | 4.0 |
| ΔR [Å] | 0.20 | 0.52 | 0.50 | 0.60 |
| ΔQ [amu$^{1/2}$Å] | 0.70 | 5.10 | 6.22 | 5.49 |
| ZFS [Ghz] | 3.42 | 2.91 | 2.46 | 3.42 |

We show the zero-phonon line (ZPL), zero field splitting (ZFS), transition dipole moment (TDM), the sum of the ionic displacements between the ground and excited state (ΔR), and the weighed sum by each ion mass (ΔQ), and radiative lifetimes.

(6.5 ns) is close to the experimental value (10 ns[33]), which can be considered as the average between the spin-selected lifetimes of 7.8 ($m_s = 0$) and 12.0 ($m_s = \pm 1$) ns[34]. Our results indicate that of all NV-like defects, the $Bi_{Ca}V_O{}^-$ complex is expected to have a bright emission in the telecommunication range (L-band 1565–1625 nm[35]).

The ΔR and ΔQ (the sum of the ionic displacements between the ground and excited states weighed by each ion mass) for the NV center agree with previously calculated values[36]. For the $X_{Ca}V_O$ defects, ΔRs are larger and ΔQs are much larger than the corresponding values in diamond, due to the size of the dopants. We find that the dopant displacement between the ground and excited state accounts for most of the ΔR values (75–85%). The large ΔQs indicate undesirable, large Huang-Rhys (HR) factors ($S_k \propto \omega_k q_k^2$ [36]), which point at small Debye-Waller factors, indicating a low quantum efficiency, i.e., the ZPL intensity is expected to be weak compared to that of the phonon sideband. Based on our results, the Debye-Waller factors for the $X_{Ca}V_O$ defects are expected to be much smaller than that of the NV center in diamond (~3%), which is not ideal for quantum technology applications. However, Debye-Waller factors may be enhanced with nanostructuring, as demonstrated for the silicon vacancy in SiC[37] where the factor was increased from 6% in bulk to 58% in nanowires. Our phonon calculations (see Supplementary Note 4 and 5) indicate that the most delocalized bulk phonons (those with an inverse participation ratio below 0.01) are responsible for the large HR factor, indicating that thin films and nanostructured CaO may have more favorable HR and DW factors and ZPL better separated from phonon sidebands (see Supplementary Fig. 5).

We find that the theoretical ZFS results for the $X_{Ca}V_O$ defects and the NV center are comparable. The calculated ground state ZFS (3.42) for the NV center in diamond is higher than the previously calculated value (2.88)[38] (due to approximations discussed in the Methodology section). We assume that this overestimation also applies to the $X_{Ca}V_O$ defects in CaO, hence we expect that the experimental ZFS of NV-like defects is likely 20% smaller compared to the value obtained in our calculations.

Experimental validation of our results should be relatively straightforward, given the ease of growth of CaO. The material has been epitaxially grown by Molecular Beam Epitaxial (MBE) or related techniques for at least two decades[39,40], although care must be exercised as samples need to be protected from moisture exposure since Ca(OH)₂ may readily form. In addition, techniques to implant Bi in CaO are available[41,42].

In summary, several simple oxides, particularly CaO, have been predicted to be promising hosts of spin defects with long coherence times. Using a high-throughput search based on first principles calculations, we predicted a class of spin defects in CaO with properties remarkably similar to those of the NV center in diamond. Such NV-like defects ($X_{Ca}V_O$) consist of a missing Ca-O pair and a dopant X=Sb, Bi, and I; they are stable, with a triplet ground state, in their negatively charged (Sb, Bi) and positively charged (I) states. They also exhibit two singlet excited states between the ground and first triplet excited states, as explicitly verified in the case of Bi, suggesting the possibility of an optical cycle similar to that of the NV center. Importantly, the $X_{Ca}V_O$ complexes have a detectable zero phonon line close to the telecommunication range and exhibit a zero-field splitting similar to the NV center in diamond. In particular, we predict that the $Bi_{Ca}V_O{}^-$ complex has a bright emission in the L-band. In addition, the presence of a high spin nucleus, strongly coupled to the electron spin, leads to many spin levels addressable in experiments and to the emergence of avoided crossings in the spin energy spectrum. We showed that when operating at these avoided crossings, the spin coherence of the $Bi_{Ca}V_O{}^-$ complex is increased by at least two orders of magnitude, exceeding seconds. In closing, the results presented in our paper pave the way to designing and engineering quantum states with long coherence times in CaO and other simple oxides.

## Methods

The defects in CaO were created using the ADAQ[17,18] software package and the the high-throughput toolkit[43]. These include vacancies, substitutionals, and interstitials, as well as pair combinations of all single defects. The maximum distance between double defects was set to 8.5 Å, which corresponds approximately to the fourth nearest neighbor distance in CaO. The dopants were limited to all non-radioactive elements in the *s*- and *p*-block of the periodic table. Using these settings, we generated 15,446 defects. We then omitted interstitial-interstitial clusters (6581), and were left with 9077 defects that were processed in the automatic screening workflow present in ADAQ, see ref. 17 for more details.

The lattice parameter of CaO was optimized with the PBE functional and found to be 4.829 Å, which is close to the experimental value of 4.811 Å[44]. We used supercells with 512 atoms for CaO and diamond. In the case of CaO, considering a dielectric constant of 11.95, we obtained a Lany-Zunger charge correction[45] of $0.058*q^2$ eV (see Supplementary Note 3A for more details). We assumed that the choice of the functional (semi-local or hybrid) and charge corrections do not affect the formation energy trend of defects obtained at the PBE level of theory.

The computations were performed with the Vienna Ab initio Simulation Package (VASP)[46,47] with the gamma compiled version 5.4.4, which uses the projector augmented wave (PAW)[48,49] method. We used three different functionals: the semi-local exchange-correlation functional of Perdew, Burke, and Erzenerhof (PBE)[24], the screened hybrid functional of Heyd, Scuseria, and Ernzerhof (HSE06)[50,51] with the standard mixing parameter α set to the standard value (25%) and (62.5%, more details in Supplementary Note 3), and the K-PBEO[52] functional (no range separation) with $\alpha = 0.29$. In ADAQ, the plane wave energy and kinetic energy cutoff of PBE calculations were set to 600 and 900 eV, respectively. For the hybrid calculations, these were reduced to 400 and 800 eV, respectively. The total energy criterion was set to $10^{-6}$ eV for PBE, and $10^{-4}$ eV in ADAQ and for the hybrid calculations. The structural minimization criterion is set to $5 \times 10^{-5}$ eV for PBE, $5 \times 10^{-3}$ eV for ADAQ, and $10^{-2}$ eV/Å for the hybrid calculations. Gaussian smearing and $\Psi_k = \Psi^*_{-k}$ are used. The pseudopotentials from VASP folder dated 2015-09-21 are Ca_pv, O, N, As_d, P,Sb, Bi_d, I, and C.

The ZPLs and absorption energies are calculated using constrained DFT (Δ-SCF)[53] for each functional. The TDMs between the ground and excited state are calculated using the wave functions from

the relaxed ground and excited state, which provide an accurate polarization and lifetime[54]. The ion relaxation between the ground and excited state are characterized by $\Delta R = \Sigma_i (R_{ex,i} - R_{gr,i})$ and $\Delta Q^2 = \Sigma_i m_i (R_{ex,i} - R_{gr,i})^2$, where $R_i$ is the ionic position and $m_i$ is the ionic mass[36]. The ZFSs are calculated using the dipole-dipole interaction of the spins[38], as implemented in VASP. We note that the method in ref. 38 uses a stand-alone code that calculates the ZFS only from the pseudo partial waves, whereas the implementation in VASP includes all contributions. The overestimate between the theory and experiment obtained using the VASP implementation for the NV center is expected to be similar for the NV-like defects in CaO.

Quantum Defect Embedding Theory (QDET) calculations are performed using the WEST (Without Empty STates) code[55]. In QDET, an active space is formed by localized defect orbitals. The normalization of the single-particle levels in the active space and the screening of the Coulomb interaction within the active space due to the remaining environment is determined from many-body perturbation theory calculations. In QDET, an effective Hamiltonian in second quantization is formulated, the diagonalization of which yields the excitations of orbitals belonging to the active space. For more methodological details, see ref. 19.

As a starting point for the QDET calculations, DFT electronic-structure calculations are performed using the Quantum Espresso code. Due to the high computational cost of QDET, calculations are performed for defected supercells with 53, 63, and 215 atoms. We have carefully analyzed the convergence both of the defect geometry and the QDET excitations with supercell size.

Nuclear spin bath-limited spin coherence was computed using the coupled cluster expansion (CCE) approach with the PyCCE package[21]. The CCE method approximates the coherence as a product of irreducible cluster contributions:

$$\mathcal{L}(t) = \prod_C \tilde{L}_C(t) = \prod_i \tilde{L}_{\{i\}}(t) \prod_{i,j} \tilde{L}_{\{ij\}}(t) \ldots \quad (1)$$

where $\tilde{L}_{\{i\}}(t)$ is the contribution of a single bath spin $i$, $\tilde{L}_{\{ij\}}(t)$ is a contribution of a spin pair $i,j$ and so on. More details are available in refs. 56,57. The CCE approach was successfully applied to the Bi-based systems before, specifically Bi donor in Si, see refs. 31,58,59. In the current work, we find that the coherence function converges at the third order of CCE with Monte Carlo bath state sampling[30] up to 2 mT away from the clock transition. The coherence functions, computed using CCE and generalized CCE (gCCE[30]), are identical in this system.

## Data availability
The ADAQ high-throughput data are available at https://httk.org/adaq/.

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

## Acknowledgements

We thank Vrindaa Somjit for many useful discussions. J.D. acknowledges support from the Swedish e-science Research Centre (SeRC), the Knut and Alice Wallenberg Foundation through the WBSQD2 project (Grant No. 2018.0071), and the Swedish Research Council (VR) Grant No. 2022-00276. M.O.acknowledges the support from a Google PhD Fellowship and a fellowship by the Qubbe center, which is supported by the National Science Foundation. C.V. and G.G. acknowledge support from the Air Force Office of Scientific Research (AFOSR) through the CFIRE grant # FA95502310667. This work used the WEST and pyCCE codes whose development is supported by MICCoM, which is part of the Computational Materials Sciences Program funded by the U.S. Department of Energy, Office of Science, Basic Energy Sciences, Materials Sciences, and Engineering Division through Argonne National Laboratory. The DFT computations were enabled by resources provided by the National Academic Infrastructure for Supercomputing in Sweden (NAISS), partially funded by the Swedish Research Council through grant agreement no. 2022-06725. The QDET calculations using the WEST code and pyCCE calculations were carried out at the RCC center at the University of Chicago.

## Author contributions

J.D. conceived the project, with support from the other authors, and performed the high-throughput and hybrid DFT calculations. C.V. performed the quantum embedding calculations. M.O. performed the coherence calculations. G.G. supervised the work. All authors discussed the results, and wrote the manuscript.

## Funding

## Competing interests

The authors declare no competing interests.
