## [Peer Review File · Nature Communications]

REVIEWER COMMENTS

Reviewer #1 (Remarks to the Author):

Joel Davidsson et. al. has used high-throughput screening method to find qubit candidate defects in CaO and further study them in detail using density functional theory, Quantum Defect Embedding theory and Cluster Correlation Expansion methods. Authors find Vacancy (Vo) + interstitial (XCa) defect complexes, where X= Sb, Bi, or I, having electronic structure apparently similar to famous NV-1 center in diamond. They further show that Hahn-echo time T2 can reach upto seconds for atomic clock-like transitions for a BiCaVO –1 defect. Authors also calculate thermodynamic and magneto-optical properties e.g. Formation energy, zero field splitting (ZFS), Zero phonon line energies, mass weighted difference in the geo-metrical co-ordinates of ground and excited states and transition dipole moments etc for these defects.

Although study present some interesting results but these are not exciting enough to be considered for publication in Nature communication. I have following criticisms and comments and authors should consider submitting their article to a specialized physics journal after addressing these comments and making these improvements.

1. The mass-weighted difference in the displacement between ground and excited state for all the proposed defects for “qubit technologies” is very large and as rightly commented by the authors in their manuscript would make the Huang-Rhys (HR) factor very large. This really makes these defects unsuitable for any qubit or ODMR related application, where high-quantum efficiency is desirable. Author do briefly mention the plausible structural engineering to overcome this limitation, but this is highly unlikely to happen in near future as there is no evidence of existence of the “studied defects” in CaO, in the first place. Moreover, intentional creation of defects of a particular type has not advanced to the level where such defects can be formed in these oxides.
2. The defects XCaVO have high formation energies so the probability of their formation and their natural concentration is likely very small. This raises scalability issues for qubit operation associated with small density of defects.
3. The authors should calculate the HR factors and PL line shapes for shortlisted defects to get an actual idea of how well separated the ZPL is from the PSB.
4. Authors mention “Our results indicate that there should be such a cycle for the NV-like defects in CaO as well”. However, for predicting a complete optical cycle of the defects both radiative and non-radiative recombination rates and inter-systems crossing rates need to be calculated. Although this is a tedious work. Authors could at least label the multi-electron levels in Fig. 4. with the symmetry

label and then use the powerful group theory to realistically establish which dipole allowed and non-radiative transitions are possible. This will help in establishing whether a particular spin-sub level of these triplets would be preferentially populated by the optical pumping as is the case for Nv-1 centre in diamond.

5. It is interesting result that although the fundamental band gap for CaO, is very different when using PBE, HSE, K-PBE0 XC functionals. But the ZPL energies for the studied transitions is very same for all the used functionals. This needs to be understood.

Minor comments/typos:

1. However, we could not obtain a satisfactory convergence of the excited states for P and As, hence in the following, “we only focus on Sb, Bi and I????”.

2. Caption of Fig. 4 “Results for diamond taken are from Ref. [19].”

3. Caption of Fig. 2 “ $10^{-4} \text{ \AA}^{3/2}$ ”.

Reviewer #2 (Remarks to the Author):

This paper uses high-throughput screening to identify a series of defects in CaO that could be used in quantum applications. The authors follow up with higher accuracy computations to characterize the defects electronic structure. CaO has been suggested as an exceptional host for quantum defects in a recent screening work. The identified defects are vacancy-substitutional complexes and among them the Bi-VacO negatively charged is the most attractive defect.

The work is interesting and definitely tackles an important question as to know what defects could be of interest in this exciting new potential host CaO. The computational methods used are for the most part state-of-the-art making most of the predictions credible. There is however an important shortcoming in the computations that cast some doubts about the possible experimental realization of this defect in the charge state needed.

The main issue is with respect of the possibility to achieve the right charge state for the targeted defect. The authors study the defect energetics versus Fermi level (see Figure 3) and indicate that CaO would need to be highly n-type for Bi-VacO to be negatively charged.

First of all, the use of a PBE functional to come to this conclusion is very dangerous. I appreciate that the screening can only be done at the PBE level but the plots in Figure 3 should all be performed with hybrids functionals that will provide a better band gap and transition levels. Hybrids will be more expensive but this is needed to reach state-of-the-art defect computations and maximum predictive power.

I will now use the PBE results in Figure 3 (which should be ideally replaced by hybrid results) to point out another even more fundamental issue. It is far from clear if a CaO material with such a n-type doping will ever be possible. Large band gap semiconductors/insulators are known to not allow certain types of doping. The numerous papers about the challenge in doping ZnO p-type are a good illustration. There is extensive literature on this problem and I would point out for instance to: Robertson, J. & Clark, S. J., Phys. Rev. B 83, 075205 (2011) and the many papers by the community on this issue. In the quantum defect community that has been also discussed for instance in Chen, Y., Turiansky, M. E. & Van De Walle, C. G., Phys. Rev. B 106 (2022) for BeO. The paper the authors cite to justify a potential n-type doping CaO does not demonstrate this. It only shows with single-particle DOS that Mo would be a shallow donor. Hole/electron compensation by intrinsic defects should be looked at to see if any attempt to dope CaO n-type could not be compensated for instance by the calcium vacancy.

On a more technical note, the authors should provide more information on the charge correction that has been used. This is critical for reproducibility. They mention the Lany-Zunger correction. Is this correction universal for any defect? Did the authors test with different supercell size if this universal correction would work? There are more recent charge corrections schemes that have emerged such as the Freysoldt/Kumagai approach that should be considered as well.

Similarly, the convergence in QDET should be provided in supplementary information.

Another technical question relates to the sentence “we could not obtain a satisfactory convergence of the excited states for P and As”. What happened there? Could the author be a bit more specific maybe with some SI data.

In several part of the papers, the authors have not given the values of the axis. This is really odd. This is the case for Fig. 3 and 2b. What are the energy values on the y for Fig. 2b and x axis for Fig. 3?

Finally, the authors claim CaO is easy to grow in the conclusion. Do they have any experimental evidence for that. Maybe a report on high quality single crystals CaO?

Reviewer #3 (Remarks to the Author):

The authors mainly predict spin defects in calcium oxide with electronic properties similar to those of the NV centers in diamond, by using high-throughput first-principles calculations. The predicted defects are charged complexes with a dopant atom — Sb, Bi, or I — occupying the volume vacated by adjacent cation and anion vacancies, of which the Bi complex emits in the telecommunication range and has a viable optical cycle for qubit initialization. Moreover, the high-spin nucleus of each dopant strongly couples to the electron spin, leading to the emergence of atomic clock-like (CT) transitions well protected from environmental noise, confirmed by cluster-correlation expansion (CCE) calculations.

From my viewpoints, the results in this manuscript are very interesting and will be important for quantum technologies with defects in solids. The newly discovered defects in oxide greatly enrich the candidate materials for hosting defects with better quantum properties. The systematic calculations in the manuscript well support the conclusions. So I recommend publication of this manuscript after the authors address the minor comments below:

- (1) It will be better if the authors add some figures to show the absorption or emission spectra of the defect complexes, to highlight the position and proportion of zero-phonon lines.
- (2) I suggest the authors briefly review the CCE method for calculating the defect coherence, by citing the relevant early references about the CCE method and also those about the CCE methods used in Bi donors in silicon.
- (3) It will be helpful to experimentalists if the author discuss more about how to form the defect complexes in oxide in practice.

Reviewer #1 (Remarks to the Author):

Joel Davidsson et. al. has used high-throughput screening method to find qubit candidate defects in CaO and further study them in detail using density functional theory, Quantum Defect Embedding theory and Cluster Correlation Expansion methods. Authors find Vacancy (Vo) + interstitial (XCa) defect complexes, where X= Sb, Bi, or I, having electronic structure apparently similar to famous NV-1 center in diamond. They further show that Hahn-echo time T2 can reach upto seconds for atomic clock-like transitions for a BiCaVO -1 defect. Authors also calculate thermodynamic and magneto-optical properties e.g. Formation energy, zero field splitting (ZFS), Zero phonon line energies, mass weighted difference in the geo-metrical co-ordinates of ground and excited states and transition dipole moments etc for these defects.

Although study present some interesting results but these are not exciting enough to be considered for publication in Nature communication. I have following criticisms and comments and authors should consider submitting their article to a specialized physics journal after addressing these comments and making these improvements.

We thank the referee for the positive comments on our interesting results. We believe the results of our paper are important and exciting, and we welcome the opportunity to emphasize them again in our response. We present robust predictions of an exciting spin defect with optically addressable clock transitions, leading to coherence times beyond seconds. This is an exciting prediction, in our opinion, as noted also by Referee 2 and 3.

To appreciate the complexity of such a prediction and the accuracy of the techniques used, we chose a promising host, as pointed out by theoretical predictions published recently in PNAS (Kanai et al. PNAS 2022). We then conducted an extensive high-throughput search, entirely from first principles, encompassing the calculations of multiple ground and excited state properties, and we additionally predicted coherence properties. Most importantly, the system discovered in our search— *a spin qubit in a simple oxide, with characteristics similar to the NV center in diamond, but with transitions in the telecom range and long coherence times* – is extremely promising. Both our computational strategy and the predictions reported in our paper will stimulate future work on spin defects in oxides; hence, our paper contains many interesting ideas for experimentalists to pursue.

1. The mass-weighted difference in the displacement between ground and excited state for all the proposed defects for “qubit technologies” is very large and as rightly commented by the authors in their manuscript would make the Huang-Rhys (HR) factor very large. This really makes these defects unsuitable for any qubit or ODMR related application, where high-quantum efficiency is desirable. Author do briefly mention the plausible structural engineering to overcome this limitation, but this is highly unlikely to happen in near future as there is no evidence of existence of the “studied defects” in CaO, in the first place. Moreover, intentional creation of defects of a particular type has not advanced to the level where such defects can be formed in these oxides.

The referee is right: our predictions have not yet been verified (but the ideas put forward are important and exciting for the community to consider and work on) and, unsurprisingly, not all problems are solved, as it most often happens when a new prediction is made and a new system is tackled. In our specific case, the Huang-Rhys (HR) factor will need to be improved either by strain engineering or most likely nanostructuring. Several groups are working on spin defects in oxide hosts, and we expect rapid progress in the near future. As an example, when the predictions of coherence times in oxides were published in PNAS 2022, soon afterward, long coherence times were measured, as predicted, in

CaWO₄ [Ourari, Salim, et al. "Coherence Properties of Single Erbium Ions in CaWO₄." *APS March Meeting Abstracts*. Vol. 2022. 2022.]

2. The defects XCaVO have high formation energies so the probability of their formation and their natural concentration is likely very small. This raises scalability issues for qubit operation associated with small density of defects.

The formation energies of the defects considered in our paper (around 10 eV at low temperature) are not significantly larger than those of other well-known defects. For example, the silicon vacancy and divacancy in SiC have formation energies of about 8 eV, and the formation energy of the tin vacancy center in diamond is around 10 eV. All these defects have been produced in sufficient quantities. Furthermore, all the defects discussed in our paper are on the defect hull, indicating that these defects are in a stable geometry and not prone to rearrange their atomic positions and attain a lower energy structure.

3. The authors should calculate the HR factors and PL line shapes for shortlisted defects to get an actual idea of how well separated the ZPL is from the PSB.

We thank the reviewer for the suggestion. We have carried out the suggested calculations for the Bi defect and added the results to the SI. The HR factor is large, around 20. However, we have identified specific bulk phonons responsible for this large value that, if eliminated, would bring the factor down to 1.11. Although it is not yet known in detail how to lock in or eliminate these unfavorable modes, our insights provide interesting avenues to pursue, e.g., by nanostructuring and working with thin films. Hence, we believe a high HR factor is not a make-or-break property.

We have added the following comment to the main text:

“

Our phonon calculations (see SI) indicate that the most delocalized bulk phonons (those with an inverse participation ratio below 0.01) are responsible for the large HR factor, indicating that thin films and nanostructured CaO may have much more favorable HR and DW factors and ZPL better separated from phonon sidebands (see Figure S6).

“

4. Authors mention “Our results indicate that there should be such a cycle for the NV-like defects in CaO as well”. However, for predicting a complete optical cycle of the defects both radiative and non-radiative recombination rates and inter-systems crossing rates need to be calculated. Although this is a tedious work. Authors could at least label the multi-electron levels in Fig. 4. with the symmetry label and then use the powerful group theory to realistically establish which dipole allowed and non-radiative transitions are possible. This will help in establishing whether a particular spin-sub level of these triplets would be preferentially populated by the optical pumping as is the case for Nv-1 centre in diamond.

We thank the reviewer for the suggestion. We have updated Fig. 4 and labeled the many body states and indicated, based on group theory, the expected splitting due to spin-orbit coupling.

5. It is interesting result that although the fundamental band gap for CaO, is very different when using PBE, HSE, K-PBE0 XC functionals. But the ZPL energies for the studied transitions is very same for all the used functionals. This needs to be understood.

The fundamental band gap values with semi-local and hybrid functionals have been obtained as *eigenvalue differences*. The ZPL values have instead been obtained as *total energy differences*, using

the Delta-SCF method. Total energy differences benefit from the cancellation of errors not present in eigenvalues differences, which are notoriously dependent on the amount of exact exchange included in the hybrid functional.

Minor comments/typos:

1. However, we could not obtain a satisfactory convergence of the excited states for P and As, hence in the following, “we only focus on Sb, Bi and I????”.

Additional details have been added to the SI.

2. Caption of Fig. 4 “Results for diamond taken are from Ref. [19].”

We corrected the caption.

3. Caption of Fig. 2 “ $10^{-4} \text{ \AA}^{3/2}$ ”.

We apologize but we did not understand the error pointed out here.

Reviewer #2 (Remarks to the Author):

This paper uses high-throughput screening to identify a series of defects in CaO that could be used in quantum applications. The authors follow up with higher accuracy computations to characterize the defects electronic structure. CaO has been suggested as an exceptional host for quantum defects in a recent screening work. The identified defects are vacancy-substitutional complexes and among them the Bi-VacO negatively charged is the most attractive defect.

The work is interesting and definitely tackles an important question as to know what defects could be of interest in this exciting new potential host CaO. The computational methods used are for the most part state-of-the-art making most of the predictions credible.

We thank the reviewer for the positive comments.

There is however an important shortcoming in the computations that cast some doubts about the possible experimental realization of this defect in the charge state needed.

The main issue is with respect of the possibility to achieve the right charge state for the targeted defect. The authors study the defect energetics versus Fermi level (see Figure 3) and indicate that CaO would need to be highly n-type for Bi-VacO to be negatively charged.

First of all, the use of a PBE functional to come to this conclusion is very dangerous. I appreciate that the screening can only be done at the PBE level but the plots in Figure 3 should all be performed with hybrids functionals that will provide a better band gap and transition levels. Hybrids will be more expensive but this is needed to reach state-of-the-art defect computations and maximum predictive power.

We thank the reviewer for the excellent suggestion. We carried out calculations with hybrid functionals, as suggested. We chose the HSE functional as it has been extensively used in the literature in studies of oxide materials, although the band gap of CaO remains underestimated with HSE.

When using the HSE06 functional, we find results qualitatively similar to PBE. However, since the band gap is increased relative to PBE, the region of stability increases for all defects. We expect that with other functionals that reproduce the measured band gap better than HSE06, the region of stability might be even greater, thus reducing the need for highly n-doped samples. However, finetuning of the functional falls outside the scope of this paper.

We added section to the SI discussing these results in detail, included in Fig. S2 our results with hybrid functionals.

I will now use the PBE results in Figure 3 (which should be ideally replaced by hybrid results) to point out another even more fundamental issue. It is far from clear if a CaO material with such a n-type doping will ever be possible. Large band gap semiconductors/insulators are known to not allow certain types of doping. The numerous papers about the challenge in doping ZnO p-type are a good illustration. There is extensive literature on this problem and I would point out for instance to: Robertson, J. & Clark, S. J., Phys. Rev. B 83, 075205 (2011) and the many papers by the community on this issue. In the quantum defect community that has been also discussed for instance in Chen, Y., Turiansky, M. E. & Van De Walle, C. G., Phys. Rev. B 106 (2022) for BeO. The paper the authors cite to justify a potential n-type doping CaO does not demonstrate this. It only shows with single-particle DOS that Mo would be a shallow donor. Hole/electron compensation by intrinsic defects should be looked at to see if any attempt to dope CaO n-type could not be compensated for instance by the calcium vacancy.

Having now obtained results with hybrid functionals, we found that the required n-doping level is not as stringent as predicted by PBE. Furthermore, regarding the negative doping with Mo, this is not the only viable route and in the revised manuscript we mention the realization of doping with a Schottky diode. Regarding the comment on spontaneous defect formation: the referee is correct that, in principle, there could be single defects that limit the allowed doping range. We overlooked this issue in the original manuscript and are grateful to the reviewer for pointing this out. We added a section to the SI that reports the formation energies of the single defects found in our ADAQ (PBE) search with the lowest formation energy. Our HSE results show that none of these defects have negative formation energy. These findings indicate that it should be possible to dope the materials with the defect of interest without forming undesired point defects.

We added the following statement to the revised manuscript:

“

We also carried out calculations with the HSE functional (see SI), showing an increased stability of the $\mathrm{X}_{\{\mathrm{Ca}\}\mathrm{V}_{\mathrm{O}}}$ defects relative to the PBE results. In addition, the stability of the isolated dopants does not constrain the value of the Fermi level required for doping, irrespective of the growth conditions of the CaO crystal (whether Ca- or O-rich), see SI.

“

On a more technical note, the authors should provide more information on the charge correction that has been used. This is critical for reproducibility. They mention the Lany-Zunger correction. Is this correction universal for any defect? Did the authors test with different supercell size if this universal correction would work? There are more recent charge corrections schemes that have emerged such as the Freysoldt/Kumagai approach that should be considered as well.

We have provided more details about the charge correction used in the SI. The Lany-Zunger (LZ) correction is used in conjunction with the PBE functional; it is the same correction regardless of the defect, and it is used for high-throughput results. At the supercell sizes (around 20 Å) used in the high-throughput calculations, the LZ charge correction is 58 meV. For the hybrid results, we also tested the FNV correction, and a comparison with the LZ correction is given in the SI. There are only minor differences between the results obtained with the two corrections for the defects studied in our work. Hence, we conclude that the changes due to the chosen charge correction to the formation energy are negligible.

Another technical question relates to the sentence “we could not obtain a satisfactory convergence of the excited states for P and As”. What happened there? Could the author be a bit more specific maybe with some SI data.

The P and As dopants have two local minima. One is located closer to the calcium site, and the other is closer to the oxygen site. Only the minimum closest to the calcium site has the same electronic structure as the other considered defects. However, the energy difference between the two minima is rather small, only in the 100 meV range. Additionally, we were unable to achieve satisfactory convergence in the excited states. Because of this, our focus shifted to the Sb, Bi, and I dopants. We have included this discussion in the SI.

In several part of the papers, the authors have not given the values of the axis. This is really odd. This is the case for Fig. 3 and 2b. What are the energy values on the y for Fig. 2b and x axis for Fig. 3?

We added the x-values for Fig. 3 and, consequently, the formation energy figure in the supplementary information. Fig 2b is a schematic picture of the defect state relative to the band gap. This figure is a standard plot for defects and was left unchanged.

Finally, the authors claim CaO is easy to grow in the conclusion. Do they have any experimental evidence for that. Maybe a report on high quality single crystals CaO?

We added the following short discussion to the conclusions, in the main text:

“

Experimental verification of our results should be relatively straightforward, given the ease of growth of CaO. The material has been epitaxially grown by MBE or related techniques for at least two decades[41,41], although care must be exercised as samples need to be protected from moisture exposure since $\text{Ca}(\text{OH})_2$ may readily form. In addition, techniques to implant Bi in CaO are available[43,44].

“

Reviewer #3 (Remarks to the Author):

The authors mainly predict spin defects in calcium oxide with electronic properties similar to those of the NV centers in diamond, by using high-throughput first-principles calculations. The predicted defects are charged complexes with a dopant atom — Sb, Bi, or I — occupying the volume vacated by adjacent cation and anion vacancies, of which the Bi complex emits in the telecommunication range and has a viable optical cycle for qubit initialization. Moreover, the high-spin nucleus of each dopant strongly couples to the electron spin, leading to the emergence of atomic clock-like (CT) transitions well protected from environmental noise, confirmed by cluster-correlation expansion (CCE) calculations.

From my viewpoints, the results in this manuscript are very interesting and will be important for quantum technologies with defects in solids. The newly discovered defects in oxide greatly enrich the candidate materials for hosting defects with better quantum properties. The systematic calculations

in the manuscript well support the conclusions. So I recommend publication of this manuscript after the authors address the minor comments below:

We thank the reviewer for the positive comments.

(1) It will be better if the authors add some figures to show the absorption or emission spectra of the defect complexes, to highlight the position and proportion of zero-phonon lines.

We have added the PL spectra to the SI and added the following comment to the main text:

“

Our phonon calculations (see SI) indicate that the most delocalized bulk phonons (those with an inverse participation ratio below 0.01) are responsible for the large HR factor, indicating that thin films and nanostructured CaO may have much more favorable HR and DW factors and ZPL better separated from phonon sidebands (see Figure S6).

“

(2) I suggest the authors briefly review the CCE method for calculating the defect coherence, by citing the relevant early references about the CCE method and also those about the CCE methods used in Bi donors in silicon.

We added the following discussion to the methodology section:

“

Nuclear spin bath-limited spin coherence was computed using the coupled cluster expansion (CCE) approach with the PyCCE package [21]. The CCE method approximates the coherence as a product of irreducible cluster contributions:

$\begin{equation}\label{eq:l_cce}$

$$\mathcal{L}(t) = \prod_{C} \tilde{L}_C(t) = \prod_{i} \tilde{L}_{\{i\}}(t) \prod_{i,j} \tilde{L}_{\{ij\}}(t) \dots$$

$\end{equation}$

where $\tilde{L}_{\{i\}}(t)$ is the contribution of a single bath spin S_i , $\tilde{L}_{\{ij\}}(t)$ is a contribution of a spin pair $S_{i,j}$ and so on. More details are available in Ref.57 and 58. The CCE approach was successfully applied to the Bi-based systems before, specifically Bi donor in Si, see Refs. 40, 59 and 60. In the current work we find that the coherence function converges at the third order of CCE with Monte Carlo bath state sampling[39] up to 2 mT away from the clock transition. The coherence functions, computed using CCE and generalized CCE (gCCE[39]), are identical in this system.

“

(3) It will be helpful to experimentalists if the author discuss more about how to form the defect complexes in oxide in practice.

We added a short discussion to the manuscript.

“

The Fermi level could also be adjusted by using a Schottky diode as e.g., in deep-level transient spectroscopy[27]. Our results indicate that if X is implanted in CaO at high T, where Schottky defects are expected to be present, NV-like defects should be formed upon annealing since the $\mathrm{X}_{\mathrm{Ca}}\mathrm{V}_{\mathrm{O}}$ cluster is more stable than the separate X_{Ca} and V_{O} point defects. We also carried out calculations with the HSE functional (see SI), showing an increased stability of the $\mathrm{X}_{\mathrm{Ca}}\mathrm{V}_{\mathrm{O}}$ defects relative to the PBE results. In addition, the stability of the isolated dopants does not constrain the value of the Fermi level required for doping, irrespective of the growth conditions of the CaO crystal (whether Ca- or O-rich), see SI.

“

REVIEWER COMMENTS

Reviewer #1 (Remarks to the Author):

The Authors have significantly improved the manuscript and have made changes as per my suggestion. I support the publication of the manuscript in its current form in Nature Comm.

Reviewer #2 (Remarks to the Author):

My main concern about the original paper was the possibility to actually stabilize the target bismuth complex in the required negative charge state. I suggested to perform hybrid calculations of formation energy vs fermi level for the complex and for intrinsic defects to look if the intrinsic defects could prevent formation of the right charged state through compensation.

The revised version unfortunately falls short of answering my concerns and I cannot recommend publication.

First, the authors use HSE with a 25% exact exchange which strongly underestimates the band gap. I am puzzled about this choice. For such a high band gap host it is expected that a higher exact exchange percentage should be used (look for instance at the paper previously published on BeO) to obtain an accurate band gap (and defect levels). The authors should have at least “tuned” the exact exchange to reproduce the experimentally known band gap of CaO as it is widely done in the field. The errors made on the band gap in the revised version are very important (a few eVs) and make the results very questionable especially for a predictive exercise.

Second, the intrinsic defects should have been computed with an adequate hybrid functional as well. The authors decided to only focus on interstitial oxygen but in any case because the hybrid functional used is not adequate for the band gap they cannot make the conclusion they are making.

Finally, the PBE reported defects in Fig. S3 are very suspicious. The authors found a calcium vacancy that is neutral on the whole range of Fermi level. This would be the only binary oxide I have ever seen that has no stable negative charge state for the cation vacancy in the gap. This is very very likely to be just wrong. This matters as calcium vacancy could be a compensating defect and by not

computing or computing wrongly the negatively charge calcium vacancy the authors give the impression this compensation cannot happen. Previous literature on CaO intrinsic defect computed with a semi local functional could have been looked at : J. Osorio-Guillén, S. Lany, S. V. Barabash, and A. Zunger “Magnetism without Magnetic Ions: Percolation, Exchange, and Formation Energies of Magnetism-Promoting Intrinsic Defects in CaO” Phys. Rev. Lett. 96, 107203. This is arguably an old paper but we can see that the calcium vacancy has a negatively charged stable defect.

Reviewer #3 (Remarks to the Author):

I think the authors have satisfactorily addressed all the concerns and comments of mine and other reviewers. So I recommend publication of the manuscript in the current form.

Response to Referees

We would like to express our gratitude to Reviewers 1 and 3 for their positive feedback and encouragement towards publication. We are pleased that Reviewer 2 was satisfied with most of our revisions, and we would like to apologize for any confusion caused by our misunderstanding of their main, previous concern. We address below in detail all the remaining issues raised by Reviewer 2.

Reviewer #1 (Remarks to the Author):

The Authors have significantly improved the manuscript and have made changes as per my suggestion. I support the publication of the manuscript in its current form in Nature Comm.

Reviewer #2 (Remarks to the Author):

My main concern about the original paper was the possibility to actually stabilize the target bismuth complex in the required negative charge state. I suggested to perform hybrid calculations of formation energy vs fermi level for the complex and for intrinsic defects to look if the intrinsic defects could prevent formation of the right charged state through compensation.

The revised version unfortunately falls short of answering my concerns and I cannot recommend publication.

First, the authors use HSE with a 25% exact exchange which strongly underestimates the band gap. I am puzzled about this choice. For such a high band gap host it is expected that a higher exact exchange percentage should be used (look for instance at the paper previously published on BeO) to obtain an accurate band gap (and defect levels). The authors should have at least “tuned” the exact exchange to reproduce the experimentally known band gap of CaO as it is widely done in the field. The errors made on the band gap in the revised version are very important (a few eVs) and make the results very questionable especially for a predictive exercise.

We thank the reviewer for clarifying their suggestion that we misunderstood in the previous round of reviews. We have now tuned the short-range Hartree-Fock exchange to reproduce the band gap, following the procedure adopted by Chen, Y., Turiansky, M. E. & Van De Walle, C. G., Phys. Rev. B 106 (2022). We report our results in the main text and section S3 of the SI.

A short-range Hartree-Fock exchange of 62.5% gives a band gap of 7.47 eV for CaO. After subtracting the zero-point renormalization (0.34 eV), we obtain a band gap of 7.13 eV, close to the experimental value of 7.09 eV. We recalculated the defect complexes in CaO using this tuned hybrid functional. For the Sb and Bi complexes, we find that the negative charge state is stable from a Fermi level of about 5 eV up to the CBM. The positive charge state of the I complex is stable from the VBM up to a Fermi level of 3 eV. See revised Figure 3 in the main text.

Second, the intrinsic defects should have been computed with an adequate hybrid functional as well. The authors decided to only focus on interstitial oxygen but in any case because the

hybrid functional used is not adequate for the band gap they cannot make the conclusion they are making.

Following the reviewer's suggestion, we calculated all single defects with the tuned HSE functional with 62.5% short-range Hartree-Fock exchange. The results have been added as a new section to the SI (Section S3) with a summary in the main text. We considered both Ca-rich and O-rich conditions. The doping levels are pinned by the oxygen and calcium vacancies. However, we find that it is possible to reach the required Fermi levels where the complexes proposed in our paper are stable without creating additional single defects. We have added a detailed discussion to section S3B, thus addressing the main concern of the reviewer.

Finally, the PBE reported defects in Fig. S3 are very suspicious. The authors found a calcium vacancy that is neutral on the whole range of Fermi level. This would be the only binary oxide I have ever seen that has no stable negative charge state for the cation vacancy in the gap. This is very very likely to be just wrong. This matters as calcium vacancy could be a compensating defect and by not computing or computing wrongly the negatively charged calcium vacancy the authors give the impression this compensation cannot happen. Previous literature on CaO intrinsic defect computed with a semi local functional could have been looked at : J. Osorio-Guillén, S. Lany, S. V. Barabash, and A. Zunger "Magnetism without Magnetic Ions: Percolation, Exchange, and Formation Energies of Magnetism-Promoting Intrinsic Defects in CaO" Phys. Rev. Lett. 96, 107203. This is arguably an old paper but we can see that the calcium vacancy has a negatively charged stable defect.

We would like to first provide some additional details on the PBE results in the original manuscript. The PBE results for the single defects (as for all other 9000 defects) are calculated by the ADAQ workflow. The software automatically selects which charge states (positive and negative) are calculated, depending on their stability. For the case of the calcium vacancy, a combination of band gap error and defect state absent from the band gap led to neglecting the calculation of the negative charge state. Thus, that charge state was not part of the results shown in Figure S3 of the original manuscript.

We agree with the reviewer that the calcium vacancy could be a compensating defect. Hence, we recalculated all single defects with the tuned HSE functional mentioned above, also adding the double charge states (double positive and double negative). The new results have been added to Section S3B, where we show that a range exists where the negative charge state of the calcium vacancy is stable and that the double negative charge state is the most stable one for higher values of the Fermi level. The results obtained with the tuned HSE qualitatively agree with those of the PRL mentioned by the referee. The difference is that in the PRL, a scissor operator is used to correct for the band gap error, leading to an unphysically small range of stability of the neutral charge state. Using the tuned HSE functional does not only increase the band gap but also lowers the VBM position, giving rise to a larger neutral stability range, as seen in Figure S3 of the revised manuscript.

The PBE results in the original Figure S3 were used to determine which defect had the lowest formation energy to reduce the number of required hybrid calculations. This is no longer needed

since all defects are now calculated at the tuned HSE level. Hence, we have removed the PBE results from the supplementary material.

Reviewer #3 (Remarks to the Author):

I think the authors have satisfactorily addressed all the concerns and comments of mine and other reviewers. So I recommend publication of the manuscript in the current form.

REVIEWERS' COMMENTS

Reviewer #2 (Remarks to the Author):

The authors addressed my concerns. The paper can be published.